# Sex Differences of Radiation Damage in High-Fat-Diet-Fed Mice and the Regulatory Effect of Melatonin

**DOI:** 10.3390/nu15010064

**Published:** 2022-12-23

**Authors:** Jingming Ren, Tong Yuan, Hang Li, Xin Wu, Junling Zhang, Deguan Li, Lu Lu, Saijun Fan

**Affiliations:** Institute of Radiation Medicine, Chinese Academy of Medical Sciences and Peking Union Medical College, Tianjin Key Laboratory of Radiation Medicine and Molecular Nuclear Medicine, Tianjin 300192, China

**Keywords:** ionizing radiation, sexual dimorphism, high-fat diets, gut microbiota, melatonin

## Abstract

The consumption of a high-fat diet (HFD) and exposure to ionizing radiation (IR) are closely associated with many diseases. To evaluate the interaction between HFDs and IR-induced injury, we gave mice whole abdominal irradiation (WAI) to examine the extent of intestinal injury under different dietary conditions. Melatonin (MLT) is a free radical scavenger that effectively prevents hematopoietic, immune, and gastrointestinal damage induced by IR. However, its effects on WAI-induced intestinal injury in HFD-fed mice remain unclear. We demonstrated that MLT can promote intestinal structural repair following WAI and enhance the regeneration capacity of Lgr5^+^ intestinal stem cells. In addition, we investigated the effects of radiation damage on sexual dimorphism in HFD-fed mice. The results showed that the degree of IR-induced intestinal injury was more severe in the HFD-fed female mice. MLT preserved the intestinal microbiota composition of HFD-fed mice and increased the abundance of Bacteroides and Proteobacteria in male and female mice, respectively. In conclusion, MLT may reduce the negative effects of HFD and IR, thereby providing assistance in preserving the structure and function of the intestine.

## 1. Introduction

Economic development and the continuous improvement of living standards have caused the human diet structure to show high fat, high calorie, and low fiber characteristics. The high-fat diet (HFD) is gradually becoming dominant. An HFD may lead to obesity, diabetes, inflammatory bowel disease, non-alcoholic fatty liver disease (NAFLD), and metabolic syndrome [1,2,3].

In addition, an HFD is closely associated with the occurrence and development of colorectal cancer, hepatocellular carcinoma, breast cancer, and other tumors. Intestinal microbiota imbalance mediated by an HFD promotes K-Ras gene mutation, damages intestinal barrier integrity, and increases inflammatory responses [4]. Through the PPAR-δ signaling pathway, an HFD enhances the proliferation and tumorigenicity of Lgr5^+^ intestinal stem cells (ISCs) [5]. HFD-induced endoplasmic reticulum stress promotes TNF-α-associated inflammatory signaling in hepatocytes, leading NAFLD to develop into hepatocellular carcinoma [6]. HFD leads to primary mammary gland tumorigenesis and lung metastasis in mice through involvement in angiogenesis [7].

Radiotherapy is one of the most important treatment modalities for tumors. With advancements in radiotherapy technology, radiotherapy has a beneficial effect on local tumor regression and prolongs the survival time of patients. Nevertheless, ionizing radiation (IR) causes oxidative stress by acting on water molecules in cells and producing large amounts of free radicals. Free radicals react with DNA and proteins, eventually leading to apoptosis and organ dysfunction [8]. Studies have shown that gastrointestinal tissues are highly sensitive to IR [9]. IR damages the crypt-villus structure of intestinal tissue, destroys intestinal barrier integrity, and affects the function of the small intestine [10]. The main clinical manifestations are nausea, vomiting, intestinal obstruction, and intestinal perforation [11]. IR-induced intestinal injury not only reduces the quality of patient survival but also interrupts the process of radiotherapy and affects the outcome of tumor treatment. Therefore, identifying an effective radioprotector is of clinical significance.

Melatonin (MLT) is a natural hormone that is produced mainly by the pineal gland [12]. MLT regulates biorhythms and exerts anti-inflammatory effects. Additionally, MLT acts as a free radical scavenger and regulates the activity of antioxidant enzymes, such as superoxide dismutase and glutathione peroxidase [13]. Studies have shown that MLT can effectively prevent hematopoietic, immune, and gastrointestinal damage induced by IR [14,15]. In addition, MLT improved HFD-induced changes in plasma proteins and inhibited the increase in fibrinogen, thereby reducing the risk of thrombosis [16]. MLT ameliorated HFD-induced liver steatosis, inflammatory responses, and gut microbiota diversity in a mouse model [17]. However, the effect of MLT on IR-induced injury in the presence of an HFD remains unclear.

Several studies have reported significant sex differences in HFD and HFD-related diseases. Compared with female mice, male mice treated with an HFD had worse glucose tolerance, inhibited increases in pancreatic β cell area, and decreased insulin sensitivity [18]. An HFD can significantly damage the immune system of male rats to a greater extent than female rats [19]. In addition, MLT is sexually dimorphic, with females possessing a higher MLT production and reserve capacity than males [20]. Therefore, we investigated whether MLT has different protective effects against whole abdominal irradiation (WAI)-induced intestinal injury in male and female mice treated with an HFD or a normal diet (ND). Our observations demonstrated that MLT can improve the structure and function of the small intestine and enhance the regenerative ability of Lgr5^+^ ISCs in irradiated mice fed an ND or HFD. Treatment with MLT restored the intestinal microbiota composition in both male and female mice fed an HFD. In summary, these data indicate that MLT may protect against WAI-induced intestinal injury and alleviate the interference of an HFD on intestinal microbiota.

## 2. Materials and Methods

### 2.1. Animals

The 6–8-week-old male and female C57BL/6 mice were purchased from H Bioscience Co. (Beijing, China). The mice were reared under sterile conditions at the Experimental Animal Center of the Institute of Radiation Medicine (IRM), Chinese Academy of Medical Sciences (CAMS). All mice were kept in a place with 12-h light-dark cycles. The mice were fed an ND or HFD for two weeks. The distribution of dietary nutrients is shown in Figure 1A. All experiments were approved by the Institutional Animal Ethics Committee of the IRM, CAMS (No. 2019096), and were performed in accordance with the National Institutes of Health Guide for the Care and Use of Laboratory Animals.

### 2.2. Irradiation and MLT Treatments

All experiments used a ^137^Cs γ-ray source placed in a Gammacell-40 irradiator with a dose rate of 0.88 Gy/min. Mice received 15 Gy of WAI, while the remainder of the body was shielded with a lead plate. The male mice were randomly divided into six groups (10 mice per group): (1) ND, (2) ND + WAI, (3) ND + WAI + MLT, (4) HFD, (5) HFD + WAI, and (6) HFD + WAI + MLT. The same was done for female mice. Mice in the ND and HFD groups were subjected to sham irradiation. For the ND + WAI + MLT and HFD + WAI + MLT groups, melatonin-containing water (0.5 mg/mL) was prepared throughout the experiment and kept in foil-wrapped bottles to protect the MLT from light degradation. The MLT dose and schedules are shown in Figure 1B. MLT consumption was performed using drinking water supplemented with MLT (Aladdin Biotech, Shanghai, China) at a dosage of 0.5 mg/mL before exposure to WAI and continued until the end of the experiment. The water-treated group (control) used the same amount of water. MLT (10 mg/kg) was administered intraperitoneally to mice in the ND + WAI + MLT and HFD + WAI + MLT groups 30 min before irradiation. The mice were euthanized five days after WAI. The spleen, thymus, liver, and inguinal white adipose tissue (iWAT) were weighed and collected to calculate the organ index. Intestinal and colon tissues were obtained for subsequent histopathology and immunohistochemistry analysis.

### 2.3. Histopathology

To assess the extent of liver and iWAT damage caused by an HFD, we resected the liver and iWAT tissues of the mice five days after irradiation. The mice were exposed with 15 Gy and sacrificed on the fifth day after WAI to evaluate intestinal injury. All collected tissues were soaked in 4% paraformaldehyde. The tissues were then embedded in paraffin and sectioned. The sections were 3–5 μm in thickness and stained with hematoxylin and eosin (H&E) and periodic acid-Schiff (PAS) as previously described [9]. Briefly, for H&E staining, the sections were successively stained with hematoxylin and eosin, then sealed with neutral values. For PAS staining, small intestine sections were stained with Schiff’s reagent for 10 min. After counterstaining with hematoxylin, the sections were dehydrated and fixed. A Pannoramic P250 digital scanner (3DHISTECH, Budapest, Hungary) was used to scan all slides, and CaseViewer2.4 software (3DHISTECH, Budapest, Hungary) was used to analyze images.

### 2.4. Immunohistochemistry Analysis

For immunohistochemistry (IHC) staining, the sections of intestinal tissues, which we collected five days after irradiation, were deparaffinized, rehydrated, and incubated with anti-Villin antibody (1:100 dilution; ab130751; Abcam, Cambridge, UK), anti-LGR5 antibody (1:200 dilution; ab75732; Abcam, Cambridge, UK), anti-lysozyme antibody (1:100 dilution; ab108508; Abcam, Cambridge, UK), and anti-Ki67 antibody (1:200 dilution; ab15580; Abcam, Cambridge, UK). The sections of colon tissues were incubated with an anti-Muc2 antibody (1:2000 dilution; ab272692; Abcam, Cambridge, UK). Then, the above sections were incubated with biotinylated secondary antibodies. Representative images of all sections were taken using a microscope (Olympus; DP26, Japan). 

### 2.5. Bacterial Diversity Analysis

To evaluate the effects of an HFD and MLT on the gut microbiota in mice, male and female mice were separately and randomly divided into three groups (six mice per group): (1) ND, (2) HFD, and (3) HFD + MLT. We obtained fresh stool samples from the mice and stored them at –80°C. DNA was extracted from stool samples using the CTAB/SDS method, according to the manufacturer’s instructions (MoBio, Carlsbad, CA, USA). PCR amplification was performed using the Phusion^®^High-Fidelity PCR Master Mix (New England Biolabs, Ipswich, MA, USA). After mixing the PCR products in equal density ratios, the QIAquick Gel Extraction Kit (Qiagen, Hilden, Germany) was used to purify the products. To evaluate bacterial diversity, the 16S ribosomal RNA (rRNA) V4 gene was analyzed using the Illumina NovaSeq platform (Novogene Bioinformatics Technology Co., Ltd., Beijing, China). The effective tags of all samples were clustered using QIIME2 software. Alpha diversity and beta diversity were calculated from the output normalized data, which were processed using QIIME2 software (Version QIIME2-202006).

### 2.6. Statistical Analysis

Data analysis was performed using GraphPad Prism version 8 software. A one-way ANOVA was used to assess the statistical significance of the differences between experimental groups. All data are expressed as mean ± standard error of the mean. The level of statistical significance was set at *p* < 0.05.

## 3. Results

### 3.1. MLT Reduces the Damage of Liver and iWAT in Mice Induced by WAI

To determine whether an HFD affects radiation-related injury, mice were randomly treated with an HFD or ND for two weeks (Figure 1B). By monitoring the body weight of mice, we found that the HFD was more likely to increase the weight of male mice than that of female mice (Figure 1C). The liver is the main site for lipid digestion, absorption, synthesis, and transport. The livers and iWAT of mice were collected and weighed. Representative images of liver tissue and iWAT are shown (Appendix A). The liver index was significantly decreased in HFD-fed male mice compared to ND-fed male mice. However, there was no significant change in the liver index of HFD-fed female mice compared to that of ND-fed female mice (Appendix A). In addition, H&E staining showed no pathological differences between male and female liver tissues from the same group (Figure 1D,E). The size of hepatocytes in the ND, ND + WAI, and ND + WAI + MLT groups was uniform, with clear nuclear structure and normal hepatic sinusoids. Intrahepatocellular lipid droplet formation was observed in the HFD and HFD + WAI + MLT groups. In the HFD + WAI group, the perisinusoidal space was slightly reduced (Figure 1D,E, black arrow).

HFD feeding markedly increased the iWAT index of male mice and decreased the iWAT index of female mice (Appendix A). As shown in Figure 1F,G, compared with the ND group, white fat was deposited, and the adipocyte area was increased in the HFD group. Both ND- and HFD-fed mice exhibited reduced adipocyte volumes after WAI. When comparing the HFD + WAI group of male mice to female mice, female mice showed more serious iWAT damage and a notable increase in adipocyte density after WAI. MLT reduced iWAT damage in irradiated mice of both sexes, regardless of an ND or HFD.

### 3.2. MLT Promotes Small Intestinal Structure Repair in Mice Following WAI

Given that the small intestine is the major site of injury during radiotherapy, representing a loss of gastrointestinal structure in mouse models, we evaluated damage to the small intestine in ND- and HFD-fed mice following 15 Gy WAI with or without MLT. H&E staining confirmed that the structure of the small intestine in the HFD group was similar to that of the ND group. The intestinal crypt villi were broken in irradiated mice (Figure 2A,B). It is worth mentioning that in female mice, small intestinal crypt-villus damage was more severe in the HFD + WAI group than in the ND + WAI group, suggesting that HFD may promote WAI-induced intestinal injury (Figure 2B). In contrast, the intestinal structure was well preserved in mice treated with MLT.

The main function of goblet cells is to synthesize and secrete mucoproteins (Muc), which form a mucosal barrier to protect intestinal epithelial cells (IECs) [21]. PAS staining showed that the number of goblet cells in the HFD and ND groups was similar in male mice, indicating that short-term HFD had no obvious effect on male mice, whereas the number of goblet cells in the HFD and ND groups in female mice was statistically different (Appendix A). Irradiation reduced the number of goblet cells in mice (Figure 2C,D). Compared with the male HFD + WAI group, the number of goblet cells in the female HFD + WAI group was expressively reduced, indicating that female mice fed a short-term HFD were less tolerant to WAI-induced intestinal injury. However, MLT treatment improved the number of goblet cells.

We further examined the expression of small intestinal villi using IHC staining. There was no discrepancy in the height of the small intestinal villi between the HFD and ND groups. After exposure to 15 Gy WAI for 5 d, the mice in the irradiated group had significantly decreased villus height compared to the sham irradiation group (Figure 2E,F). Compared with WAI-treated mice, those treated with MLT exhibited increased villus height (Appendix A). Collectively, these results suggest that MLT ameliorates the small intestinal injury caused by WAI.

### 3.3. MLT Enhances the Regenerative Ability of Lgr5^+^ ISCs after WAI

IECs have the ability to continuously self-renew [22]. After destruction by external factors, epithelial cells can rapidly regenerate and restore homeostasis in intestinal cells. As a marker of intestinal stem cells, Lgr5 plays a crucial role in intestinal regeneration after radiation injury [23]. Paneth cells are IECs that secrete antibacterial proteins such as lysozyme [24]. Paneth cells participate in the maintenance of intestinal environmental homeostasis by killing intestinal microorganisms. Ki67 is a marker for regenerating epithelial cells and can be identified as proliferating cells in the small intestine [25]. Therefore, we examined the effect of MLT on WAI-induced intestinal injury using IHC staining. For both male and female mice, the results showed that the number of Lgr5^+^ ISCs and lysozyme^+^ Paneth cells in the HFD and ND groups were similar, suggesting that HFD may have no obvious effect on the expression of these cells (Figure 3A–D and Appendix A). However, the number of Ki67^+^ cells in the HFD group of female mice was lower than that in the ND group (Appendix A). Compared with the sham irradiation group, the number of Lgr5^+^ ISCs, lysozyme^+^ Paneth cells, and Ki67^+^ cells was significantly reduced in the irradiated group five days after WAI, although these changes were reversed after MLT treatment (Figure 3A–F and Appendix A). These results illustrate that MLT can improve the regenerative capacity of ISCs and sustain homeostasis in the intestine by enhancing the proliferation and differentiation of crypt cells.

### 3.4. MLT Alleviates WAI Induced Colon Injury

To further explore the severity of intestinal injury caused by WAI in mice, we collected the colons of the mice and measured their length after irradiation. As shown in Appendix A, the colons of mice were shortened after WAI. The colon length of MLT-treated mice was similar to that of sham-irradiated mice. Notably, an HFD shortened the colon length in female mice but not in male mice (Figure 4A,E). Compared to the ND group, H&E staining showed leukocyte infiltration in the submucosa of the colon in the HFD group, and the crypt depth decreased (Figure 4B,F). After WAI, the crypt structure was partially missing, the crypt became shallow, and the glands were misaligned. Thus, MLT treatment may alleviate colonic injury.

Muc2 is a mucoprotein secreted by IECs. It can infiltrate and protect the intestinal epithelium [26]. Muc2 expression was detected using IHC staining. MLT treatment inhibited a reduction in Muc2 protein levels (Figure 4C,G). Nevertheless, the expression of Muc2 protein in HFD-fed male mice was significantly reduced compared with that in the ND group (Figure 4D,H). The above results suggest that an HFD induces sexual dimorphism in colon injury in male and female mice.

### 3.5. MLT Treatment Retains the Intestinal Bacterial Flora Composition Pattern Impaired by HFD

Studies have shown that an HFD can cause dysregulation of the gut microbiota [27,28]. We used 16S rRNA gene sequencing to study the gut microbiota of HFD-fed mice, with or without MLT treatment. High-throughput sequencing analysis showed that an HFD increased α-diversity of intestinal microbes in both male and female mice. The Chao1 index confirmed that the abundance of microbiota in the HFD-fed mice was significantly higher than that in the ND group (Figure 5A,B). However, MLT treatment restored the microbiota abundance to the approximate levels seen in the ND group. The observed operational taxonomic units (OTUs) and dominance indices also confirmed these results (Appendix A). The Simpson index indicated that the microbiota diversity of the HFD-fed mice increased (Figure 5B,F). The same phenomenon was observed using the Shannon index (Appendix A). Pielou’s evenness index showed that the species in the HFD group were uniform, while MLT treatment reduced the diversity of microbiota and improved species evenness (Figure 5C,G). Principal coordinate analysis (PCoA) and non-metric multidimensional scaling (NMDS) showed that the composition of the intestinal microbiota was clearly separated after HFD and MLT treatment in male and female mice, suggesting that HFD and MLT shaped the intestinal microbiota profile. To distinguish the differentially abundant OTUs, we used linear discriminant analysis (LDA) and effect size (LEfSe) analyses to show the differences in microbiota abundance among the three groups of mice. Compared with the ND group, Clostridia and Firmicutes were more abundant in the HFD group of male mice, while Acidobacteriales were more abundant in the HFD group of female mice (Figure 5I–L). After MLT treatment, the abundance of Bacteroides and Proteobacteria in male and female mice, respectively, increased. These data demonstrate that MLT may restore the intestinal microbiota structure altered by an HFD.

### 3.6. MLT Treatment Restores the Changes in Gut Microbiota Abundance Caused by an HFD

At the genus level, community analysis among the three groups showed that the genus with the highest relative abundance was Muribaculaceae. In male mice, HFD treatment increased the relative abundance of Blautia (Figure 6A). HFD treatment diminished the relative abundance of Lactobacillus in female mice (Figure 6B). MLT treatment restored the abundance of the corresponding microbiota to levels close to those present in the ND group.

## 4. Discussion

MLT is an indoleamine neuroendocrine hormone secreted by the pineal gland, which has various physiological effects on circadian rhythm regulation, antioxidant expression, and anti-inflammatory activity [12]. Owing to its wide range of physiological activities and reliable safety, MLT is added to many health foods to improve human health in Europe and the United States [29]. MLT is amphiphilic; its lipophilicity allows it to act across cell and nuclear membranes, and its hydrophilicity allows it to act in the intercellular space [30]. As a free radical scavenger, MLT can concentrate in free radical-rich mitochondria and protect lipids, proteins, and DNA from oxidative stress damage [31]. Studies have shown that MLT can transfer electrons from blood cells and repair DNA damage [32]. In addition, MLT modulates the activities of several antioxidant enzymes, such as superoxide dismutase, glutathione peroxidase, and catalase [13].

With continuous progress in science and technology, radiotherapy has become an important clinical therapeutic method. IR causes damage to the body through direct and indirect action [8]. Direct action refers to the direct damage of proteins and DNA by IR. Indirect action means that IR acts on water molecules to produce a large number of free radicals, which further damage biological macromolecules and eventually lead to cell apoptosis. However, IR not only targets lesions but also causes damage to normal tissues and induces a series of adverse reactions, including nausea, anemia, and skin irritation [11]. A previous study showed that gastrointestinal tissue is very sensitive to IR [9]. IR-induced acute intestinal injury is characterized by the death of crypt epithelial cells and the inflammation of the lamina propria [33]. Therefore, it is extremely important to identify remedial measures to protect organisms from radiation-induced damage. Studies have shown that MLT can effectively prevent hematopoietic, immune, and gastrointestinal damages induced by IR [14,15].

In addition to the intestinal injury induced by IR, an HFD can also cause damage to the intestines by affecting intestinal microbiota homeostasis, intestinal barrier function, and the intestinal immune system. An HFD regulates the expression of the tight junction-associated ZO-1 protein and increases levels of the pro-inflammatory factors TNF-α, IL-1β, and IL-6, thereby causing intestinal inflammation [1]. Previous studies have shown that an HFD causes oxidative stress and endoplasmic reticulum stress in IECs, damages the mucosal barrier, and reduces the number of goblet cells and Muc2 protein levels in mice [34]. Free fatty acids derived from an HFD also cause atrophy of gut-associated lymphoid tissue and decrease the number of small intestinal epithelial and lamina propria lymphocytes, disrupting the intestinal immune barrier [35]. By affecting the maturation of B cells, an HFD reduces IgA antibodies and accordingly breaks down intestinal resistance to pathogenic substances [36].

There are limited reports on the effects of an HFD on IR-induced injury. Yoshida et al. showed that IR-induced carcinogenesis in C3H mice is significantly reduced after caloric restriction [37]. An HFD notably enhances the efficacy of radiotherapy in malignant glioma mice [38]. Obesity increases oxidative stress induced by IR [39]. Nevertheless, the effect of MLT on IR-induced injury in the presence of an HFD remains elusive. Thus, we first fed mice an ND and HFD for two weeks and then administered 15 Gy WAI to determine whether MLT could ameliorate IR-induced intestinal injury. The results indicated that MLT treatment promoted the repair of crypt-villus structures and increased the number of goblet cells in irradiated small intestinal tissues (Figure 2). Lgr5 is a marker of ISCs, and Lgr5^+^ ISCs are essential for intestinal regeneration after exposure to IR [23]. Paneth cells secrete lysozymes. Lgr5^+^ ISCs and Paneth cells are responsible for homeostasis in the small intestine [24]. Ki67 is a marker for the regeneration of epithelial cells [25]. IHC staining of Lgr5^+^ ISCs, lysozyme^+^ Paneth cells, and Ki67^+^ cells revealed that MLT treatment increased the levels of these cells in irradiated mice (Figure 3). Therefore, we conclude that MLT can alleviate damage to the small intestine induced by WAI and maintain the regenerative capacity of IECs. For WAI-induced colon injury, we measured the colon length of mice in the different treatment groups, and the results showed that the colon length of mice treated with MLT was similar to that of the sham irradiation group (Figure 4A,E). MLT treatment promoted crypt repair in the colon after irradiation and inhibited Muc2 protein level reduction caused by WAI (Figure 4). Furthermore, we found that 15 Gy WAI did not cause damage to the liver, but affected the volume of adipocytes, which manifested that MLT could reduce the serious damage to iWAT in mice (Figure 1D–G). The hematopoietic system of ND-fed mice was slightly damaged after WAI, whereas the hematopoietic function of HFD-fed mice was not affected (Appendix A).

Sexual dimorphism means that males and females have obvious differences in their morphological structure and often show different physiological and functional parameters. Several studies have shown significant sex differences in HFD- and HFD-related diseases. Compared to female mice, male mice are more prone to HFD-induced metabolic disorders, which are reflected in insulin resistance and weight gain [18,27]. Studies on HFD-induced liver damage have found that females have better control of hepatic mitochondrial respiratory coupling and a protective effect on hepatic steatosis and fibrosis [40]. Sex hormones have been reported to be involved in regulating HFD-induced obesity in mice [41]. In addition, MLT is associated with the secretion of estrogen and androgen via regulation of the hypothalamic-pituitary-gonadal axis [42]. Tan et al. reported that MLT is also sexually dimorphic, with females possessing a higher MLT production and reserve capacity than males [20]. Therefore, we considered whether MLT had different effects on male and female mice administered an HFD or ND. We found that an HFD seems to affect the liver and intestinal tissue of male and female mice, respectively. The liver index of HFD-fed male mice meaningfully decreased (Appendix A). HFD-fed female mice had reduced numbers of goblet cells and Ki67^+^ cells in the small intestine, as well as shorter colon length (Figure 2, Figure 3 and Figure 4). Furthermore, an HFD may promote IR-induced intestinal injury in female mice, mainly manifesting as more severe damage to the crypt-villus structure in the HFD + WAI group compared to that in the ND + WAI group (Figure 2B). The number of goblet cells in the female HFD + WAI group was significantly reduced, indicating that the female mice were less tolerant to IR-induced intestinal injury (Figure 2C,D).

An HFD is closely associated with dysbiosis of the gut microbiota, which is an important factor in the regulation of host metabolism [43]. Rats fed an HFD may have dysregulation of bile acid metabolism, which in turn inhibits the propagation of Bacteroidetes and Firmicutes and reduces the composition and diversity of the gut microbiota [44,45]. With the prolongation of the HFD feeding time, the number of harmful bacteria, Enterobacteriaceae, gradually increased, while the number of beneficial bacteria, Lactobacillus and Bifidobacterium, decreased. Cani et al. indicated that an HFD significantly increased the abundance of gram-negative bacteria in the intestine [46]. Lipopolysaccharides stimulate Toll-like receptors, causing or aggravating intestinal permeability, and disrupting the intestinal mucosal barrier [1,47]. Recently, gut microbiota have been identified as an effective target for regulating abnormal lipid metabolism [17,48]. Therefore, we used 16S rRNA gene sequencing to study the intestinal bacterial composition in HFD- and MLT-treated mice. The results showed that HFD treatment increased the relative abundance of Blautia in male mice but decreased the relative abundance of Lactobacillus in female mice at the genus level (Figure 6). Compared with ND-fed mice, MLT treatment preserved the intestinal microbiota composition of HFD-fed mice, which increased the abundance of Bacteroides in male mice and Proteobacteria in female mice (Figure 5I–L).

## 5. Conclusions

In this study, our data indicated that MLT may promote intestinal structure repair following WAI and enhance the regenerative ability of Lgr5^+^ ISCs in ND- or HFD-fed mice. We also found that female mice fed an HFD were less tolerant to IR-induced intestinal injury. An HFD has different effects on the structure and diversity of gut microbiota in male and female mice, whereas MLT ameliorates the disturbance of intestinal microbiota induced by an HFD. In summary, MLT not only plays a part in radiation protection but also alleviates the negative effects of an HFD.

## Figures and Tables

**Figure 1 nutrients-15-00064-f001:**
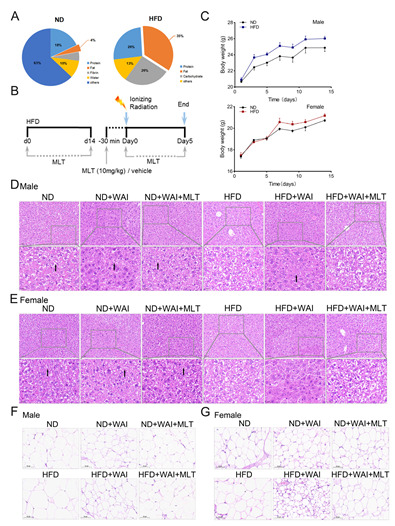
MLT reduces the damage of iWAT in mice induced by WAI. Mice were fed an HFD for 2 weeks, then given whole abdominal irradiation. MLT was injected intraperitoneally 30 min before WAI. Control mice received sham irradiation (**B**). (**A**) The fat content of a normal diet (ND) was 4%, whereas that of an HFD was 35%. (**C**) Body weights of ND-fed and HFD-fed mice. Representative H&E staining of hepatic lipids from male (**D**) and female (**E**) mice (20 × magnification); the black arrows indicate the perisinusoidal space. Representative H&E staining of iWAT from male (**F**) and female (**G**) mice (20 × magnification). (WAI, whole abdominal irradiation; HFD, high-fat diet; MLT, melatonin; H&E, hematoxylin and eosin).

**Figure 2 nutrients-15-00064-f002:**
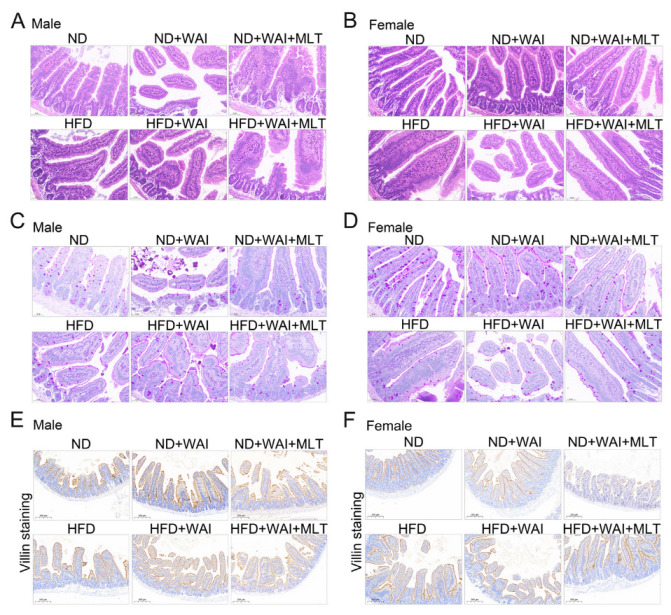
MLT promotes small intestinal structure repair in mice following WAI. Representative H&E staining of small intestinal structures from male (**A**) and female (**B**) mice (20 × magnification). Representative PAS staining of small intestinal structures from male (**C**) and female (**D**) mice (20 × magnification). Representative immunohistochemical images of villi^+^ in the small intestinal structure from male (**E**) and female (**F**) mice (10 × magnification). (MLT, melatonin; H&E, hematoxylin and eosin; PAS, periodic acid-Schiff; WAI, whole abdominal irradiation).

**Figure 3 nutrients-15-00064-f003:**
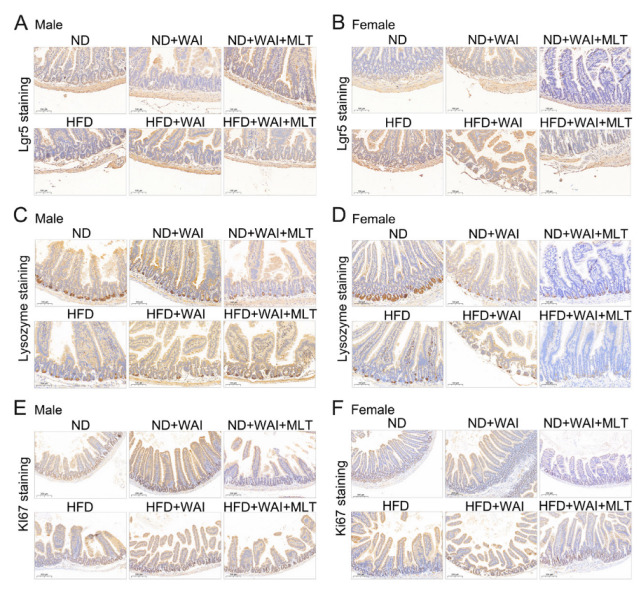
MLT enhances the regenerative ability of Lgr5^+^ ISCs after WAI. Representative immunohistochemistry images showing the expression of Lgr5^+^ (**A**), lysozyme^+^ (**C**), and Ki67^+^ (**E**) in small intestinal sections from male mice. Representative immunohistochemistry images showing the expression of Lgr5^+^ (**B**), lysozyme^+^ (**D**), and Ki67^+^ (**F**) in small intestinal sections from female mice. (10 × magnification). (MLT, melatonin; WAI, whole abdominal irradiation).

**Figure 4 nutrients-15-00064-f004:**
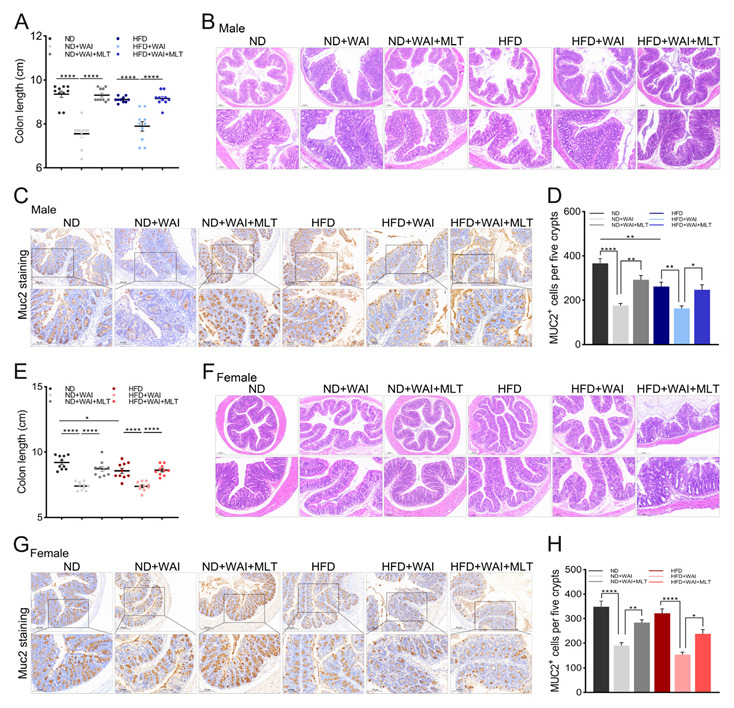
MLT alleviates WAI-induced colon injuries. (**A**) The length of colon tissues from male mice in the six groups. (**B**) Representative H&E staining of colon tissues from male mice (top: 5 × magnification; bottom: 10 × magnification). (**C**) Representative immunohistochemistry images showing the expression of Muc2^+^ in colon tissues from male mice (top: 5 × magnification; bottom: 10 × magnification). (**D**) The number of Muc2^+^ cells of male mice per image was quantified. (**E**) The length of colon tissues from female mice in the six groups. (**F**) Representative H&E staining of colon tissues from female mice (top: 5 × magnification; bottom: 10 × magnification). (**G**) Representative immunohistochemistry images showing the expression of Muc2^+^ in colon tissues from female mice (top: 5 × magnification; bottom: 10 × magnification). (**H**) The number of Muc2^+^ cells in female mice per image was quantified. The results are shown as the mean ± SEM, * *p* < 0.05, ** *p* < 0.01, **** *p* < 0.0001 between the two cohorts (5 views per group). (MLT, melatonin; WAI, whole abdominal irradiation; H&E, hematoxylin and eosin; SEM, standard error of mean).

**Figure 5 nutrients-15-00064-f005:**
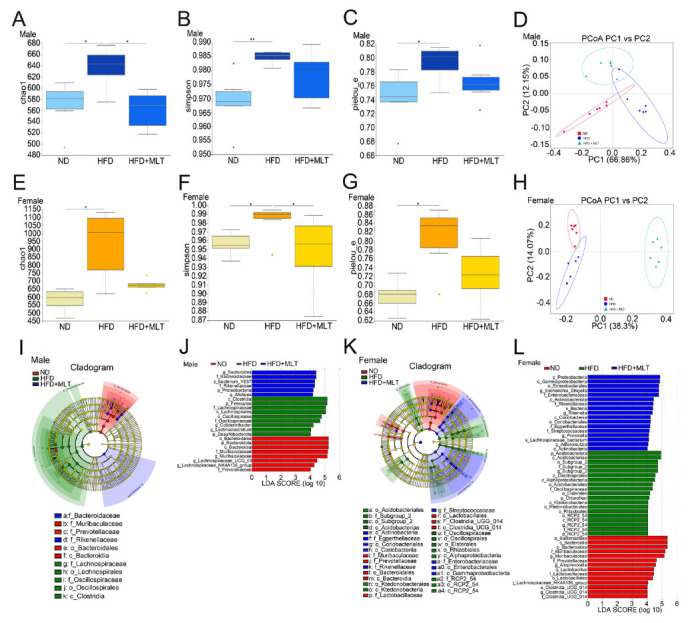
MLT treatment retains the intestinal bacterial flora composition pattern impaired by an HFD. After treating mice with an HFD and MLT for two weeks, feces were collected and evaluated by 16S rRNA high-throughput sequencing. The Chao1 diversity, Simpson, and Pielou’s evenness indices of enteric bacteria in male (**A**–**C**) and female (**E**–**G**) mice were measured. The Wilcoxon rank-sum test revealed statistical significance. PCoA was used to assess the gut microbiome taxonomic profile of male (**D**) and female (**H**) mice (*n* = 6 mice per group). LEfSe results showed that bacterial abundance was significantly different between the HFD and ND groups of male (**I**) and female mice (**K**). The red, green, and blue regions represent enriched clades in the ND, HFD, and HFD + MLT groups. The yellow regions indicate species with no significant difference, and the size of the region indicates the amount of species abundance. Microbial taxa with significant differences between the three groups of male (**J**) and female (**L**) mice were counted using LDA analysis. The results are shown as the mean ± SEM, * *p* < 0.05, ** *p* < 0.01 between the two cohorts.(MLT, melatonin; HFD, high-fat diet; PCoA, principal coordinate analysis; LEfSe, linear discriminant analysis and effect size; ND, normal diet; LDA, linear discriminant analysis).

**Figure 6 nutrients-15-00064-f006:**
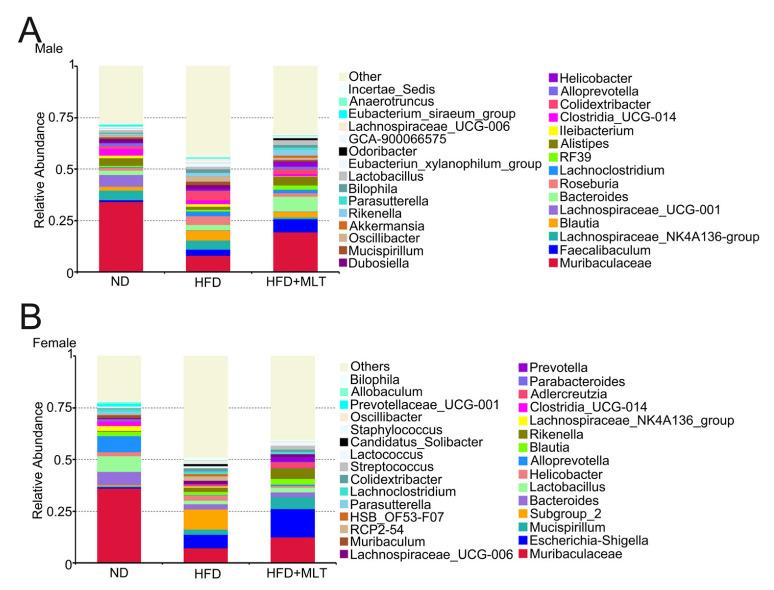
MLT treatment restores the changes in gut microbiota abundance caused by an HFD. A community analysis of the three groups was conducted by analyzing the relative abundance of intestinal bacterial flora at the genus level in mice. In the HFD group, the relative abundance of Blautia increased in male mice (**A**), while the relative abundance of Lactobacillus decreased significantly in female mice (**B**). (MLT, melatonin; HFD, high-fat diet).

## Data Availability

Datasets presented in this study are available on request from the corresponding author.

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
