# Peer review of "Sex Differences of Radiation Damage in High-Fat-Diet-Fed Mice and the Regulatory Effect of Melatonin"

_nutrients, 2022, doi:10.3390/nu15010064_

Round 1

Reviewer 1 Report

Dear Authors,

I think your work is well written and it is interesting for the scientific community. However, it needs extensive revision for a better comprehension of the quality of the results. You reports a lot of data but they have to be supported by a detailed description of the animal groups (number, sex and time and type of treatments and sample collection and analyses) for a more easy reading and to support statistically the results.

The above reported is the main important point, following some other suggestions:

- PAS staining is not reported in the Material and Methods: it has to be added.

- Housing conditions of the mice have to be added together with light-dark cycle.

- Which is the methods for MLT administration: drinking water or injection?

- What does this sentence mean: "Speen, thymus, liver and inguinal white adipose tissue (WAT) were weight and collected. Intestinal and colon tissues were obtained for subsequent experiments" (Lines 97-98). Which are the subsequent experiments?

- For the immunohistochemical analysis, the revelation system has to be specified.

- Major enlargement of microscopical image should be added to show the particular structure, (for example, where is the perisinusoidal space (line 152) visible?

Author Response

Point 1: I think your work is well written and it is interesting for the scientific community. However, it needs extensive revision for a better comprehension of the quality of the results. You reports a lot of data but they have to be supported by a detailed description of the animal groups (number, sex and time and type of treatments and sample collection and analyses) for a more easy reading and to support statistically the results.

 Response 1: We thank the reviewer for pointing out this. We have revised “Materials and Methods”, the number of mice in each group in Materials and Methods has been revised in Line 88. The gender of mice was described in the 87, 89 and 126 lines. According to your comments, we have made supplementary explanations to "Histopathology" and "Immunohistochemistry analysis" in Materials and Methods, clarifying the time and processing method of sample collection.

Point 2: PAS staining is not reported in the Material and Methods: it has to be added.

Response 2: We have supplemented the specific operating methods for PAS staining in the Materials and Methods section (Lines 109-112).

Point 3: Housing conditions of the mice have to be added together with light-dark cycle.

Response 3: We have added a supplementary description of the environment in which the mice were housed (line 79).

Point 4: Which is the methods for MLT administration: drinking water or injection?

Response 4: Administration of MLT-containing daily drinking water and intraperitoneal injection of MLT to mice before irradiation were both our methods for MLT administration. As shown in Figure 1B, for both ND+WAI+MLT and HFD+WAI+MLT groups of female and male mice, we continued to administer 0.5 mg/mL MLT throughout the experiment. And 30 min before irradiation, the above mice were injected intraperitoneally with 10 mg/mL MLT. In order to clarify the method of administration, we provide supplementary descriptions in lines 90 and 97.

Point 5: What does this sentence mean: "Speen, thymus, liver and inguinal white adipose tissue (WAT) were weight and collected. Intestinal and colon tissues were obtained for subsequent experiments" (Lines 97-98). Which are the subsequent experiments?

Response 5: Thank you for pointing out this. We have corrected this sentence, in lines 99-101.

“Spleen, thymus, liver, and inguinal white adipose tissue (iWAT) were weighed and collected to calculate the organ index. Intestinal and colon tissues were obtained for subsequent histopathology and immunohistochemistry analysis.”

Point 6: For the immunohistochemical analysis, the revelation system has to be specified.

Response 6: As suggested by the reviewer, we have specified the revelation system for IHC analysis in line 122. “A Pannoramic P250 digital scanner (3DHISTECH, Budapest, Hungary) was used to scanned all slides and CaseViewer2.4 software (3DHISTECH, Hungary) was used to analyze images.”

Point 7: Major enlargement of microscopical image should be added to show the particular structure, (for example, where is the perisinusoidal space (line 152) visible?

Response 7: We have added major enlargement of microscopical image. And arrows indicated the location of the perisinusoidal space.

Reviewer 2 Report

In the manuscript “Sex differences of radiation damage in high-fat-diet-fed mice and the regulatory effect of melatonin”, Ren et al reported how melatonin (MLT) can regulate several physiological effects. MLT is an indoleamine neuroendocrine hormone secreted by the pineal gland and is crucial for circadian rhythm regulation and has antioxidant and anti-inflammatory activities. In this study, the authors found more physiological effects of MLT: MLT can reduce the damage of liver in mice induced by whole abdominal irradiation (WAI); MLT can promote small intestinal structure repair in mice following WAI; MLT can enhance the regenerative ability of Lgr5+ ISCs after WAI; and MLT can alleviate WAI induced colon injury. In addition, MLT can preserve the intestinal microbiota composition of high-fat-diet fed mice and increase the abundance of Bacteroides and Proteobacteria in male and female mice, respectively. Therefore, these MLT treatments perhaps produce different effects dependent on the use of male or female mice. This paper is interesting and useful for further medical applications. However, some sentences must be deleted before publication.

Page 4, Line 135-137 should be deleted.

Page 9, Line 311-314 should be deleted.

Author Response

Point: In the manuscript “Sex differences of radiation damage in high-fat-diet-fed mice and the regulatory effect of melatonin”, Ren et al reported how melatonin (MLT) can regulate several physiological effects. MLT is an indoleamine neuroendocrine hormone secreted by the pineal gland and is crucial for circadian rhythm regulation and has antioxidant and anti-inflammatory activities. In this study, the authors found more physiological effects of MLT: MLT can reduce the damage of liver in mice induced by whole abdominal irradiation (WAI); MLT can promote small intestinal structure repair in mice following WAI; MLT can enhance the regenerative ability of Lgr5 ISCs after WAI; and MLT can alleviate WAI induced colon injury. In addition, MLT can preserve the intestinal microbiota composition of high-fat-diet fed mice and increase the abundance of Bacteroides and Proteobacteria in male and female mice, respectively. Therefore, these MLT treatments perhaps produce different effects dependent on the use of male or female mice. This paper is interesting and useful for further medical applications. However, some sentences must be deleted before publication.

Page 4, Line 135-137 should be deleted.

Page 9, Line 311-314 should be deleted.

Response: Thank you very much your comments. We have deleted the sentences in lines 135-137 and 311-314.